# Lifting the Information Ratio:
# An Information-Theoretic Analysis of
# Thompson Sampling for Contextual Bandits

**Gergely Neu**
Universitat Pompeu Fabra
Barcelona, Spain
gergely.neu@gmail.com

**Julia Olkhovskaya**
Vrije Universiteit Amsterdam
Amsterdam, the Netherlands
julia.olkhovskaya@gmail.com

**Matteo Papini**
Universitat Pompeu Fabra
Barcelona, Spain
matteo.papini@upf.edu

**Ludovic Schwartz**
Universitat Pompeu Fabra
Barcelona, Spain
ludovic.v.schwartz76@gmail.com

## Abstract

We study the Bayesian regret of the renowned Thompson Sampling algorithm in contextual bandits with binary losses and adversarially-selected contexts. We adapt the information-theoretic perspective of Russo and Van Roy [2016] to the contextual setting by considering a lifted version of the information ratio defined in terms of the unknown model parameter instead of the optimal action or optimal policy as done in previous works on the same setting. This allows us to bound the regret in terms of the entropy of the prior distribution through a remarkably simple proof, and with no structural assumptions on the likelihood or the prior. The extension to priors with infinite entropy only requires a Lipschitz assumption on the log-likelihood. An interesting special case is that of logistic bandits with $d$-dimensional parameters, $K$ actions, and Lipschitz logits, for which we provide a $\widetilde{O}(\sqrt{dKT})$ regret upper-bound that does not depend on the smallest slope of the sigmoid link function.

## 1 Introduction

Thompson sampling is one of the most popular algorithms for sequential decision making under uncertainty. First proposed by Thompson [1933], it has been rediscovered several times over the consequent decades and has been eventually popularized in the machine learning literature by Chapelle and Li [2011] and Scott [2010], who pointed out its excellent empirical performance for solving contextual bandit problems. These empirical studies were followed by a sequence of breakthroughs on the front of theoretical analysis, spearheaded by the works of Agrawal and Goyal [2012, 2013a,b], Kaufmann et al. [2012], and Russo and Van Roy [2014], Russo and Van Roy [2016]. Thanks to these successes, Thompson sampling has become one of the gold-standard methods for solving multi-armed bandit problems. Indeed, in the last decade, several Thompson-sampling-style methods have been developed and analyzed for a variety of problem settings.

The variety of different analysis techniques applied to Thompson sampling is perhaps even larger than the variety of problem settings that it has been applied to. The first key tools for analyzing the Bayesian regret of Thompson sampling for multi-armed bandits have been developed by Russo and Van Roy [2014], and our analysis naturally borrows several of these tools. The worst-case results developed in said work were refined in [Russo and Van Roy, 2016], where they proved for the first

36th Conference on Neural Information Processing Systems (NeurIPS 2022).

time "information-theoretic" bounds on the regret of TS that scale with the Shannon entropy of the optimal action $A^*$ under the prior on the model parameters. This result has inspired a range of follow-up works, including extensions to uncountable action sets [Dong and Van Roy, 2018], approximate implementations [Lu and Van Roy, 2017, Qin et al., 2022], and even new algorithms based on the analysis technique itself [Russo and Van Roy, 2018, Kirschner and Krause, 2018]. One limitation that this technique could not overcome so far is not being able to satisfyingly deal with *context*.

When considering i.i.d. contexts and finite policy classes, one can apply the theory of Russo and Van Roy [2016] and treat policies as actions (as done by Liu et al. 2018, Bubeck and Sellke 2020), or adapt the information-ratio directly (as done by Kirschner et al. 2020, Hao et al. 2022) to obtain regret bounds scaling with the entropy of the optimal policy. This approach, however, can lead to a polynomial dependence on the number of contexts when assuming no further structure about the rewards, as the entropy of the optimal policy may be as large as $\log |\Pi| = |\mathcal{X}| \log |\mathcal{A}|$. Another variant of Thompson sampling demonstrating a similar prior-dependent regret bound has been proposed by Li [2013], whose regret guarantees also suffer from a suboptimal dependence on the number of rounds $T$. A much more satisfying solution has been recently given by Zhang [2021], whose "feel-good Thompson sampling" method guarantees both frequentist and Bayesian regret bounds of the order $\sqrt{KT \log N}$, where $K$ is the number of actions, $T$ the number of rounds, and $N$ the support size of the prior distribution on model parameters $\theta^\star$. Under a Lipschitzness assumption on the prior and the likelihood, Zhang proves a frequentist regret bound scaling with the log-prior-probability mass assigned to the true parameter $\theta^\star$. The techniques involved in proving these results drew substantial inspiration from the works of Foster and Rakhlin [2020], Foster and Krishnamurthy [2021] and in fact results of a similar flavor were also proved recently in Foster et al. [2021].

Our own approach can be seen as a reconciliation of the analysis style of Zhang [2021] with the information-theoretic methodology of Russo and Van Roy [2016]. Our main conceptual contribution is proposing an adjustment of the now-classic notion of "information ratio" proposed by Russo and Van Roy [2016] that applies to contextual bandits. In its original definition, the information ratio quantifies the tradeoff between incurring low regret and gaining information about the optimal action. As we will argue, this notion of information gain is inappropriate for contextual bandits with non-i.i.d. contexts. We propose a variant that measures the amount of information gained about the true model parameter $\theta^\star$ instead of the optimal action (which may be context dependent). The complexity notion resulting from this extension is called the "lifted information ratio". Our analysis shows that the Bayesian regret of Thompson sampling can be bounded in terms of the lifted information ratio and the Shannon entropy of the hidden parameter $\theta^\star$, which mirrors the result of Russo and Van Roy [2016]. Along the way, we draw inspiration from the recently proposed analysis technique of Zhang [2021] for contextual bandits, and in fact we show that our notion of lifted information ratio bridges the concept of "decoupling coefficient" proposed by Zhang with the information ratio of Russo and Van Roy.

We state our main results in the context of $K$-armed contextual bandits with binary losses. For countable parameter spaces, we prove that the Bayesian regret of Thompson sampling satisfies a bound of order $\sqrt{KTH(\theta^*)}$, where $H$ denotes the Shannon entropy. This result is comparable to the bound of Foster and Krishnamurthy [2021] for the FastCB algorithm, which is of the order $\sqrt{KT \log |\Theta|}$ and holds in a frequentist sense. This is the best result we are aware of for this setting. To demonstrate the flexibility of our technique, we provide an extension to logistic bandits with Lipschitz continuous logits, generalizing the well-studied setting involving logits that are linear functions of the context and the parameter $\theta$. For this setting, we prove a regret bound of order $\sqrt{KT \log \mathcal{N}_{1/CT}(\Theta, \|\cdot\|)}$, where $\mathcal{N}_\varepsilon(\Theta, \|\cdot\|)$ is the $\varepsilon$-covering number of $\Theta$ under norm $\|\cdot\|$ and $C$ is the Lipschitz constant of the logits. This implies a regret bound of order $\sqrt{KdT}$ in the well-studied case of linear logits. Notably, the bound does not show any dependence on the smallest slope $\kappa$ of the sigmoid link function that almost all existing results for this setting suffer from [Filippi et al., 2010, Kveton et al., 2020, Abeille et al., 2021, Faury et al., 2022]. Indeed, this constant has plagued all regret bounds since the early work of Filippi et al. [2010] and was only recently moved to lower-order terms by the breakthrough work of Abeille et al. [2021]. Bounds involving other potentially large problem-

dependent constants have also been proved in the Bayesian setting by Dong and Van Roy [2018] and Dong et al. [2019]. To our knowledge, our bounds are the first to entirely remove this factor.[1]

The rest of the paper is organized as follows. After introducing the necessary technical background in Section 2, we discuss matters of information gains, information ratios, and decoupling coefficients in Section 3. We state our main results and instantiate them in a variety of settings in Section 4. We provide the key ideas of the analysis in Section 5 and conclude in Section 6.

**Notation.** For a natural number $n \in \mathbb{N}$, $[n] = \{1, 2, \ldots, n\}$ denotes the set of the first $n$ natural numbers. For $x, y \in \mathbb{R}^d$, $\langle x, y \rangle$ denotes the canonical scalar product of $x$ and $y$, and $\|x\|$ the Euclidean norm of $x$. We denote the Shannon entropy of a discrete random variable $Z$ with probability mass function $P$ as $H(Z) = -\mathbb{E}\left[\log(P(Z))\right]$. We use $\bar{0}_d$ for a $d$-dimensional vector of zeros and $I_d$ for the $d \times d$ identity matrix.

## 2   Preliminaries

We consider a parametric class of contextual bandits with parameter space $\Theta$, context space $\mathcal{X}$, and $K$ actions. To each parameter $\theta \in \Theta$ there corresponds a contextual bandit with loss distribution $P_{\theta,x,a}$ for each context $x \in \mathcal{X}$ and action $a \in \mathcal{A}$, with the mean of the loss distribution denoted by $\ell(\theta, x, a)$. We will dedicate special attention to the case where the losses are binary and thus $P_{\theta,x,a}$ is a Bernoulli distribution with parameter $\ell(\theta, x, a)$. For the main part of our theoretical analysis, we will assume $\Theta$ is either a finite set or a bounded metric space.

We study the problem of regret minimization in the Bayesian setting. In this setting, the environment secretly samples a parameter $\theta^\star$ from a known prior distribution $Q_1$ over $\Theta$. We assume that the agent has full knowledge of the prior and the likelihood model $P_{\theta,x,a}$. The agent interacts with the environment for $T$ rounds as follows. At each round $t \in [T]$, an adaptive adversary selects a context $X_t$, possibly using randomization and taking into account the previous history of actions and losses, but not $\theta^\star$. The latter is a common assumption [e.g., Agrawal and Goyal, 2013b]. After observing the context $X_t$, the agent selects an action $A_t \in \mathcal{A}$ (possibly using randomization) and incurs a binary loss $L_t \sim P_{\theta^\star, X_t, A_t}$. The goal of the agent is to minimize the expected sum of losses. In the Bayesian setting, this is equivalent to minimizing the Bayesian regret, defined as follows:

$$R_T = \mathbb{E}\left[\sum_{t=1}^{T}\left(\ell(\theta^\star, X_t, A_t) - \ell(\theta^\star, X_t, A_t^\star)\right)\right], \tag{1}$$

where $A_t^\star = \arg\min_{a \in \mathcal{A}} \ell(\theta^\star, X_t, a)$ is the optimal action for round $t$, and the expectation in (1) is over all sources of randomness: the initial sampling of $\theta^\star$ from $Q_1$, the agent's randomization over actions, and the randomness of the loss realizations.

Furthermore, let $\mathcal{F}_t = \sigma(X_1, A_1, L_1, \ldots, X_t, A_t, L_t)$ be the sigma-algebra representing the history of contexts, actions and losses observed by the agent up to time $t$ included. We use $Q_t$ to denote the distribution of the unknown parameter $\theta^\star$ conditional on the past history $\mathcal{F}_{t-1}$, and simply call it the posterior distribution. We denote by $\pi_t(\cdot|X_t)$ the distribution over the agent's actions conditional on $X_t$ and $\mathcal{F}_{t-1}$, and call it the agent's policy. Finally, we will frequently use the shorthand notations $\mathbb{E}_t\left[\cdot\right] = \mathbb{E}\left[\cdot\,|\mathcal{F}_{t-1}, X_t\right]$ and $\mathbb{P}_t\left[\cdot\right] = \mathbb{P}\left[\cdot\,|\mathcal{F}_{t-1}, X_t\right]$.

This paper is dedicated to the study of the celebrated Thompson Sampling (TS) algorithm, defined as follows. At each round $t$, TS draws a parameter $\theta_t$ from the posterior distribution $Q_t$. Then, it selects the action that minimizes $\ell(\theta_t, X_t, \cdot)$. Finally, it updates $Q_t$ via Bayes' rule, obtaining the new posterior $Q_{t+1}$. The algorithm can be equivalently defined as a method that plays actions according to their posterior probability of being optimal, that is: $\mathbb{P}\left[A_t = a|\mathcal{F}_{t-1}, X_t\right] = \mathbb{P}\left[A_t^\star = a|\mathcal{F}_{t-1}, X_t\right]$. The pseudocode is shown as Algorithm 1.

## 3   Regret, information ratio, and decoupling coefficient

The classic results of Russo and Van Roy [2016] establish that the regret of Thompson sampling in non-contextual multi-armed bandit problems can be upper bounded in terms of a quantity called the

---

[1]Despite our best efforts, we could not verify how the bounds of Zhang [2021] scale with problem-dependent factors in this setting, due to the heavy use of asymptotic notation in their proofs.

---

**Algorithm 1** Thompson Sampling (TS)

---
1: **Input:** prior $Q_1$
2: **for** $t = 1, 2, \ldots, T$ **do**
3:     Sample $\theta_t \sim Q_t$
4:     Observe context $X_t$
5:     Play action $A_t = \arg\min_{a \in \mathcal{A}} \ell(\theta_t, X_t, a)$
6:     Incur loss $L_t$
7:     Update $Q_t$ with $(X_t, A_t, L_t)$ to obtain $Q_{t+1}$ by Bayes' rule
8: **end for**

---

*information ratio.* Informally, the information ratio measures the tradeoff between achieving low regret and gaining information about the identity of the optimal action $A^\star$ (which is a deterministic function of $\theta^\star$ in the standard multi-armed bandit setting). The formal definition is given by

$$\rho_t^* = \frac{(\mathbb{E}_t \left[ \ell(\theta^\star, A_t) - \ell(\theta^\star, A^\star) \right])^2}{I_t(A_t^\star; (A_t, L_t))},$$

where $I_t(A_t^\star; (A_t, L_t))$ denotes the mutual information between $A^\star$ and the action-observation pair $(L_t, A_t)$, conditioned on the history $\mathcal{F}_{t-1}$. Intuitively, having small information ratio implies that every time Thompson sampling suffers large regret, it has to gain a lot of information about the optimal action, which suggests that it should be possible to bound the total regret by the total amount of information that there is to be gained. The result of Russo and Van Roy [2016] confirms this intuition by showing that the regret of Thompson sampling is of the order $\sqrt{\rho^* T H(A^\star)}$, where $\rho^* = \max_t \rho_t^*$ and $H$ is the Shannon entropy. The information ratio itself can always be upper bounded by $K/2$, but better bounds can be shown when the loss function has favorable structural properties (e.g., whenever the reward function is a $d$-dimensional linear function, the information ratio is at most $d/2$).

While this result and the underlying information-theoretic framework is very elegant, it is inappropriate for studying contextual bandit problems. The specific challenge is that the optimal action $A_t^\star$ changes from round to round and gaining large amount of information about $A_t^\star$ for any given round may not necessarily be useful for predicting future actions. To see this, consider a stylized example with action set $\{1, 2, \ldots, K\}$, where there exists an action $a_{\mathrm{reveal}} = 1$ whose loss entirely reveals the identity of the optimal action $a^\star(x)$ for context $x$: $\ell(\theta^\star, x, a_{\mathrm{reveal}}) = 1 - 10^{-a^\star(x)}$. Taking this action provides large information gain about $A_t^\star$, but results in large regret and reveals nothing about the future losses. Thus, in the contextual setting, one can keep following a policy that provides low information ratio while suffering linear regret. This issue necessitates an alternative definition that still permits an effective information-theoretic analysis.

Our proposition is to consider a relaxed definition of the information ratio based on the mutual information between the *true parameter* $\theta^\star$ and the observed loss. In particular, we define

$$\rho_t = \frac{(\mathbb{E}_t \left[ \ell(\theta^\star, X_t, A_t) - \ell(\theta^\star, X_t, A_t^\star) \right])^2}{I_t(\theta^\star; L_t)}, \tag{2}$$

where $I_t(\theta^\star; L_t) = \mathbb{E}_t \left[ \mathcal{D}_{\mathrm{KL}} \left( P_{L_t|\theta^\star, \mathcal{F}_{t-1}, X_t, A_t} \big\| P_{L_t|\mathcal{F}_{t-1}, X_t, A_t} \right) \right]$ is the mutual information between $\theta^\star$ and $L_t$, conditioned on the history $\mathcal{F}_{t-1}$ and the context-action pair $X_t, A_t$. This quantity measures the information that the agent gains about $\theta^\star$. High values of $I_t(\theta^\star; L_t)$ intuitively allow making better predictions about the future loss realizations for all possible context sequences. Since $A_t^\star$ is a deterministic function of $\theta^\star$ given $X_t$, the data processing inequality implies that the information gain about $\theta^\star$ is always greater than that about $A_t^\star$, which in turn implies that $\rho_t$ is smaller than what one would obtain by directly generalizing the definition of Russo and Van Roy [2016]. As this notion of information gain measures the efficiency of inferring the identity of a hidden parameter, we refer to $I_t(\theta^\star; L_t)$ as the *lifted information gain*, and $\rho_t$ as defined in Equation (2) as the lifted information ratio. As our analysis will establish, a bounded lifted information ratio guarantees low regret, and we will show that the ratio itself can be bounded reasonably under conditions similar to the ones required by the analysis of Russo and Van Roy [2016], despite the fact that all quantities are defined conditionally on a sequence of non-i.i.d. contexts..

We are not the first to consider the lifted information ratio: very similar quantities have been considered in the literature on *information-directed sampling*, as initiated by Russo and Van Roy

[2018], who have already introduced an the non-contextual counterpart of the same quantity in their Section 9.2. The recent works of Kirschner et al. [2020], Hao et al. [2022] also consider versions of the standard information ratio for contextual bandits, but their analysis crucially relies on the assumption that the contexts are i.i.d. Similarly, Lu et al. [2021], Liu et al. [2022] define versions of the information ratio using various *targets* for information-directed sampling, which recovers our definition of the lifted information ratio when setting the target as $\theta^\star$. While these works do not distinguish between all these versions of the information ratio on the level of terminology, we believe that emphasizing the "lifted" nature of our information ratio provides more clarity in that it indicates the specific features of contextual bandit learning with parametric losses in a transparent way. Thus, we will keep using this terminology throughout the paper.

Our lifted information ratio is also closely related to a quantity appearing in the analysis of Zhang [2021], called the "decoupling coefficient". Adapted to our Bayesian setting, this coefficient can be defined as the smallest constant $\delta$ such that the following inequality holds:

$$\mathbb{E}_t\left[\ell(\theta^\star, X_t, A_t) - \ell(\theta^\star, X_t, A_t^\star)\right] \leq \inf_{\mu > 0} \mathbb{E}_t\left[\mu\left(\ell(\theta^\star, X_t, A_t) - \mathbb{E}_t\left[\ell(\theta^\star, X_t, A_t)\mid X_t, A_t\right]\right)^2 + \frac{\delta}{\mu}\right]$$

$$= 2\sqrt{\delta\mathbb{E}_t\left[\left(\ell(\theta^\star, X_t, A_t) - \mathbb{E}_t\left[\ell(\theta^\star, X_t, A_t)\mid X_t, A_t\right]\right)^2\right]},$$

where the first line gives the original definition mirroring that of Zhang [2021] and the second line plugs in the choice of $\mu$ achieving the infimum. Reordering gives the value of the optimal $\delta_t^*$:

$$\delta_t^* = \frac{\left(\mathbb{E}_t\left[\ell(\theta^*, X_t, A_t) - \ell(\theta^*, X_t, A_t^*)\right]\right)^2}{4\mathbb{E}_t\left[\left(\ell(\theta^*, X_t, A_t) - \mathbb{E}_t\left[\ell(\theta^*, X_t, A_t)\mid X_t, A_t\right]\right)^2\right]},$$

which matches our definition of the lifted information ratio, up to the difference of replacing the mutual information by the root mean-squared error in predicting the true parameter $\theta^*$. Notably, this definition essentially coincides with the lifted information ratio for the special case of Gaussian losses.

## 4  Main results

In this section, we state our main results concerning the Bayesian regret of Thompson sampling for contextual bandits. We will assume that the losses are binary and the action space is finite, unless otherwise stated. However, several of our results can be generalized beyond this setting. We will illustrate this in Section 4.3, where we provide some additional results for the classic setting of Gaussian linear contextual bandits.

We begin by stating two general regret bounds in terms of the lifted information ratio defined in Equation (2). The reader that is not interested in the full generality of our theory may skip to Section 4.2 for concrete regret bounds. Our first abstract bound applies to priors with finite entropy, the simplest example being finite parameter spaces.

**Theorem 1.** *Assume $Q_1$ is supported on the countable set $\Theta_1 \subseteq \Theta$ and that the lifted information ratio for all rounds $t$ satisfies $\rho_t \leq \rho$ for some $\rho > 0$. Then, the Bayesian regret of* TS *after $T$ rounds can be bounded as*
$$R_T \leq \sqrt{\rho T H(\theta^\star)}.$$
*In particular if $\Theta_1$ is a finite set with $|\Theta_1| = N$, the regret of* TS *satisfies*
$$R_T \leq \sqrt{\rho T \log N}.$$

The proof of this theorem is stated in Section 5.1. Unfortunately, the Shannon entropy can be unbounded for distributions with infinite support, which is in fact the typical situation that one encouters in practice. To address this concern, we develop a more general result, that holds for a broader family of distributions. In the following, $(\Theta, \varrho)$ is a metric space with metric $\varrho : \Theta^2 \to \mathbb{R}$. We make the following regularity assumption on the likelihood function $P_{\theta, x, a}$:

**Assumption 1.** *There exists a constant $C > 0$ such that for any $\theta, \theta' \in \Theta_1$, $|\log P_{\theta, x, a}(L) - \log P_{\theta', x, a}(L)| \leq C\varrho(\theta, \theta')$ holds for all $x \in \mathcal{X}$, $a \in \mathcal{A}$, and $L \in \{0, 1\}$.*

Under this assumption, we can state a variant of Theorem 1 that applies to metric parameter spaces:

**Theorem 2.** *Assume $(\Theta, \varrho)$ is a metric space, and $Q_1$ is supported on $\Theta_1 \subseteq \Theta$ with $\varepsilon$-covering number $\mathcal{N}_\varepsilon(\Theta_1, \varrho)$. Let Assumption 1 hold, and assume the lifted information ratio for all rounds $t$ satisfies $\rho_t \leq \rho$ for some $\rho > 0$. Then, the Bayesian regret of* TS *after $T$ rounds can be bounded as*

$$R_T \leq \sqrt{\rho T \min_\varepsilon \left(\log\left(\mathcal{N}_\varepsilon(\Theta_1, \varrho)\right) + 2\varepsilon C T\right)}.$$

The proof is based on a covering argument on top of the proof of the previous theorem, and is provided in Appendix A.2. To get a better understanding of Assumption 1, it is useful to notice that it is satisfied in basic settings, like logistic bandits with Lipschitz logits. See Section 4.2 for details.

### 4.1   Bounding the lifted information ratio

At this point, some readers may worry that the lifted information ratio may be impossible to bound due to the lifting to the space of parameters $\Theta$. To address this concern, we now turn to showing bounds on the lifted information ratio. We first consider the unstructured case, that holds for arbitrary parameter spaces, likelihoods, and priors.

**Lemma 1.** *Suppose that the losses are binary and $|\mathcal{A}| = K$. Then, the lifted information ratio of Thompson sampling satisfies $\rho_t \leq 2K$ for all $t \geq 1$.*

The proof of this lemma (provided in Section 5.2) relies on a decoupling argument between the choice of the action and of the parameters at round $t$, inspired by Zhang [2021]. Taking his argument one step further, we center our analysis around an application of convex conjugacy, which we believe may be applicable in a broader variety of settings. We wish to highlight that this proof technique is very different from the information-theoretic methodology pioneered by Russo and Van Roy [2016].

Next, we consider the case of linear expected losses in Euclidean parameter spaces, which, in principle, allows for an unbounded number of actions.

**Lemma 2.** *Suppose that $\Theta \subseteq \mathbb{R}^d$, the losses are binary, and the expected losses are linear functions of the form $\ell(\theta, x, a) = \langle \theta, \varphi(x, a) \rangle$, where $\varphi : \mathcal{X} \times \mathcal{A} \to \mathbb{R}^d$ is a feature map, such that $\ell(\theta, x, a) \in [0, 1]$ for all $x \in \mathcal{X}, a \in \mathcal{A}, \theta \in \Theta$. Then, the lifted information ratio satisfies $\rho_t \leq d/2$.*

The proof of this result (deferred to Appendix B) follows the arguments of Proposition 5 of Russo and Van Roy [2016], with some small but important changes to account for the presence of contexts.

Notably, both of these results match the classic bounds of Russo and Van Roy [2016] on the standard definition of the information ratio for these settings (cf. their Propositions 5 and 3), implying that lifting to the space of parameters does not substantially impact the regret-information tradeoff.

### 4.2   Concrete regret bounds for Bernoulli bandits

We now instantiate our bounds in two well-studied settings for Bernoulli bandits. We start from the fully unstructured case, assuming finite actions and finitely supported prior. The following regret bound follows directly from Theorem 1 and Lemma 1.

**Theorem 3.** *Consider a contextual bandit with $K$ actions and binary losses, and suppose $\Theta_1$, the support of $Q_1$, is finite with $|\Theta_1| = N$. Then, the Bayesian regret of* TS *satisfies:*

$$R_T \leq \sqrt{2KT \log N}.$$

This result is comparable to the best known regret guarantees for this problem due to Foster and Krishnamurthy [2021] and Zhang [2021], and matches the minimax rate for unstructured contextual bandits with a policy class of size $N$ [Beygelzimer et al., 2011, Dudík et al., 2011]. However, we are not aware of a comparable result for the Thompson sampling algorithm in the literature, be it Bayesian or not.

Moving to Bernoulli bandits with structure, we consider a well-studied setting known as *logistic bandits*. In this model, the losses are generated by a Bernoulli distribution as $L_t(\theta, x, a) \sim \text{Ber}(\sigma(f_\theta(x, a)))$, where $\sigma(z) = 1/(1 + e^{-z})$ is the sigmoid function. We just assume that $f_\theta(x, a)$, called the logit function, is $C$-Lipschitz in $\theta$, which directly implies that Assumption 1 holds. Notice that our definition generalizes the commonly used notion of logistic bandits that consider linear logit functions of the form $f_\theta(x, a) = \langle \theta, \varphi(x, a) \rangle$, where $\varphi : \mathcal{X} \times \mathcal{A} \to \mathbb{R}^d$ is some feature map. Our result for logistic bandits is based on Theorem 2 and Lemma 1.

**Theorem 4.** *Assume $\Theta \subset \mathbb{R}^d$ and $\|\theta\| \le S$ for all $\theta \in \Theta_1$. Consider a class of logistic bandits with $K$ actions and $C$-Lipschitz logit function $f_\theta(x, a)$. Then, the Bayesian regret of* TS *after $T$ rounds can be bounded as:*

$$R_T \le \sqrt{2KT \left(d \log(2SCT + 1) + 1\right)}.$$

The proof follows from an application of Theorem 2, and can be found in Appendix A.3. To our knowledge, this is the first regret bound for logistic bandits with nonlinear logits.

We can further specialize our result to linear logits, the setting that is most commonly studied in the literature:

**Corollary 1.** *Assume $\Theta \subset \mathbb{R}^d$, $\|\theta\| \le S$ for all $\theta \in \Theta_1$. Consider a class of logistic bandits with $K$ actions and linear logit function $f_\theta(x, a) = \langle \varphi(x, a), \theta \rangle$, with $\|\phi(x, a)\| \le B$ for all $x \in \mathcal{X}$ and $a \in [K]$. Then, the Bayesian regret of* TS *after $T$ rounds can be bounded as:*

$$R_T \le \sqrt{2KT \left(d \log(2SBT + 1) + 1\right)} = \widetilde{O}\big(\sqrt{dKT}\big).$$

The proof of the corollary is stated at Appendix A.3. A remarkable feature of this bound is that it shows no dependence on the minimum derivative of the sigmoid link function, albeit at the price of a $\sqrt{K}$ factor in the bound. Nevertheless, we believe this to be the first regret guarantee that entirely gets rid of this potentially enormous constant without very strong assumptions. Indeed, this constant has been present in nearly all previous bounds we are aware of [Filippi et al., 2010, Li et al., 2017, Faury et al., 2020, Abeille et al., 2021, Faury et al., 2022]—although these results have the advantage of holding in a frequentist sense. In the Bayesian setting, the works of Dong and Van Roy [2018] and Dong et al. [2019] have proved a variety of bounds on the regret of Thompson sampling for non-contextual logistic bandits, but none of them are directly comparable with our result above. Dong et al. [2019] prove a regret bound of order $d\sqrt{T}$ for a highly specialized setting with $\mathcal{A} = \Theta$, and a range of other bounds under a variety of strong assumptions.

An improved bound for the many-actions setting that scales at most logarithmically with $K$ remains an open problem. Its difficulty is testified by a set of negative examples provided by Dong et al. [2019], and by a long-lived conjecture of [Dong and Van Roy, 2018] regarding the information ratio for logistic bandits, that, to our knowledge, has not yet been verified in theory.

### 4.3 Beyond binary losses: linear bandits with Gaussian noise

We now illustrate how our techniques (in particular the lifted information ratio) can be extended beyond the case of binary losses, and in particular consider the classic setting of Bayesian linear contextual bandits, where the loss is a linear function of a $d$-dimensional feature map with additive Gaussian noise, and the prior is also Gaussian.

**Lemma 3.** *Suppose $\theta^*$ be a $d$-dimensional normal random vector, so $\theta^* \sim \mathcal{N}\left(\bar{0}_d, \lambda I_d\right)$ for some $\lambda > 0$, the loss is a normal random variable $L_t \sim \mathcal{N}(\ell(\theta^\star, X_t, A_t), \sigma^2)$ for some standard deviation $\sigma > 0$, and the expected loss is linear, so $\ell(\theta, x, a) = \langle \theta, \varphi(x, a) \rangle$, where $\varphi : \mathcal{X} \times [K] \to \mathbb{R}^d$ is a feature map. Then, the lifted information ratio satisfies $\rho_t \le \min\{d, 2(1 + \log K)\}$.*

For this setting, the bound $\rho_t \le d$ has already been shown for the classic information ratio by Russo and Van Roy [2016]. However, we believe that our bound $\rho_t \le 2(1 + \log K)$ is new for any definition of information ratio. By combining this bound on the lifted information ratio with standard arguments for linear contextual bandits, we recover both of the well-known regret bounds of order $d\sqrt{T \log T}$ and $\sqrt{dT \log (KT)}$ for this seting, respectively due to Abbasi-Yadkori et al. [2011] and Chu et al. [2011]. [2] See Corollary 2 in Appendix B for a rigorous statement and the proof. This result, although not surprising, indicates once again that our notion of lifted information ratio does not lead to compromises in performance, even when the losses are not binary.

## 5 Analysis

This section presents the key ideas of the proofs of our main results. We will just provide the proof of Theorem 1 and that of Lemma 1, which we believe offer the most insight into our techniques. All

---

[2] While for the cases when $d \ll \log K$ the dependence on $K$ is not present in the regret bound, our analysis is restricted to the setting with the finite number of actions. Still, using a standard discretization argument, it is possible to extend the analysis to infinite action spaces.

other proofs, included those of auxiliary lemmas, are deferred to Appendix A and B. For sake of clarity, we focus on the relatively simple case where $\Theta_1$ is countable, so that (with a slight abuse of notation) we can write $Q_t(\theta)$ to denote the posterior probability associated with $\theta$. Note, however, that our full proofs also handle the case of general distributions (details in Appendix A.1).

## 5.1 The proof of Theorem 1

Recalling the definition of the lifted information ratio (Equation 2), we first notice that the regret can be rewritten as follows:

$$
\mathbb{E}\left[\sum_{t=1}^{T}\left(\ell(\theta^\star, X_t, A_t) - \ell(\theta^\star, X_t, A_t^\star)\right)\right] = \mathbb{E}\left[\sum_{t=1}^{T}\mathbb{E}_t\left[\ell(\theta^\star, X_t, A_t) - \ell(\theta^\star, X_t, A_t^\star)\right]\right]
$$

$$
= \mathbb{E}\left[\sum_{t=1}^{T}\sqrt{\rho_t I_t(\theta^\star; L_t)}\right] \le \sqrt{\mathbb{E}\left[\sum_{t=1}^{T}\rho_t\right]\cdot\mathbb{E}\left[\sum_{t=1}^{T}I_t(\theta^\star; L_t)\right]}, \tag{3}
$$

where the first step uses the tower rule of expectation, the second step the definition of $\rho_t$, and the final step follows from the Cauchy–Schwarz inequality.

The key challenge is then to bound the sum of information-gain terms. The following lemma provides a more tractable form of this sum:

**Lemma 4.** *Under the assumptions of Theorem 1,*

$$
\mathbb{E}\left[\sum_{t=1}^{T}I_t(\theta^\star; L_t)\right] = \mathbb{E}\left[\log\frac{\prod_{t=1}^{T}p_{\theta^\star, t}(L_t)}{\sum_\theta Q_1(\theta)\prod_{t=1}^{T}p_{\theta, t}(L_t)}\right].
$$

The proof of this lemma is based on a classic "Bayesian telescoping" argument that we have learned from Grünwald [2012]. We provide the proof of Lemma 4 in Appendix A.1. Supposing now that the prior has bounded entropy, we can easily bound the term appearing on the right hand side as follows:

$$
\mathbb{E}\left[\log\frac{\prod_{t=1}^{T}p_{\theta^\star, t}(L_t)}{\sum_\theta Q_1(\theta)\prod_{t=1}^{T}p_{\theta, t}(L_t)}\right] \le \mathbb{E}\left[\log\frac{\prod_{t=1}^{T}p_{\theta^\star, t}(L_t)}{Q_1(\theta^\star)\prod_{t=1}^{T}p_{\theta^\star, t}(L_t)}\right] = \mathbb{E}\left[-\log Q_1(\theta^\star)\right] = H(\theta^\star).
$$

This concludes the proof of the first statement. The second statement follows from the first using the trivial bound on the Shannon entropy of any finite-support distribution.

## 5.2 The proof of Lemma 1

We start by introducing some notation that will be useful for the proof. In particular, we use $g$ to denote the binary relative entropy function defined for all $p, q \in [0, 1]$ as

$$
g(p\|q) = p\log\frac{p}{q} + (1 - p)\log\frac{1 - p}{1 - q}, \tag{4}
$$

and we use the convention $0 \cdot \log 0 = 0$. Furthermore, we define the posterior mean loss as $\bar{\ell}_t(x, a) = \mathbb{E}_t\left[\ell(\theta^\star, x, a)\right]$. These notations allow us to conveniently rewrite the information gain as

$$
I_t(\theta^\star; L_t) = \mathbb{E}_t\left[\mathcal{D}_{\mathrm{KL}}\left(P_{L_t|\theta^\star, \mathcal{F}_{t-1}, X_t, A_t}\big\|P_{L_t|\mathcal{F}_{t-1}, X_t, A_t}\right)\right] = \mathbb{E}_t\left[g\left(\ell(\theta^\star, X_t, A_t)\big\|\bar{\ell}_t(X_t, A_t)\right)\right].
$$

We will now prove a generalization of Lemma 1, which will directly imply the original result:

**Lemma 5.** *Under the assumptions of Lemma 1, for all $t$, the lifted information ratio of Thompson sampling satisfies $\rho_t \le 2\sum_{a\in\mathcal{A}}\mathbb{E}_t\left[\bar{\ell}_t(X_t, a)\right]$.*

*Proof.* The proof is based on an application of the Fenchel–Young inequality, which requires the introduction of the Legendre–Fenchel conjugate of $g$ with respect to its first argument. This function is defined for all $u \in \mathbb{R}$ as

$$
g^*(u\|q) = \sup_{p\in[0,1]}\{pu - g(p\|q)\} = \log(1 + q(e^u - 1)) \le q\left(u + \frac{u^2}{2}\right), \tag{5}
$$

where the second equality and the inequality follow from a set of straightforward calculations deferred to Appendix A.4. Turning to the actual proof, we consider the instantaneous pseudo-regret $r_t = \ell(\theta^\star X_t, A_t) - \ell(\theta^\star, X_t, A_t^\star)$ in a fixed round $t$ and write the following (for any $\eta > 0$):

$$
\begin{aligned}
\mathbb{E}_t\left[r_t\right] &= \mathbb{E}_t\left[\ell(\theta^\star, X_t, A_t) - \ell(\theta^\star, X_t, A_t^\star)\right] \\
&= \mathbb{E}_t\left[\ell(\theta^\star, X_t, A_t) - \ell(\theta_t, X_t, A_t)\right] \\
&\qquad \text{(using that } (\theta^\star, A_t^\star) \text{ has the same conditional distribution as } (\theta_t, A_t)) \\
&= \mathbb{E}_t\left[\bar\ell_t(X_t, A_t) - \ell(\theta_t, X_t, A_t)\right] \\
&\qquad \text{(by conditional independence of } \theta^\star \text{ and } A_t) \\
&= \mathbb{E}_t\left[\bar\ell_t(X_t, A_t) - \sum_{a\in\mathcal{A}}\mathbb{I}_{\{A_t=a\}}\frac{\eta\pi_t(a|X_t)}{\eta\pi_t(a|X_t)}\ell(\theta_t, X_t, a)\right] \\
&\leq \mathbb{E}_t\left[\bar\ell_t(X_t, A_t) + \eta\sum_{a\in\mathcal{A}}\pi_t(a|X_t)\Big(g(\ell(\theta_t, X_t, a)\|\bar\ell(X_t, a)) \right. \\
&\qquad \left. + g^*\left(-\frac{\mathbb{I}_{\{A_t=a\}}}{\eta\pi_t(a|X_t)}\Big\|\bar\ell(X_t, a)\right)\Big)\right] \\
&\qquad \text{(by the Fenchel-Young inequality)} \\
&\leq \mathbb{E}_t\left[\bar\ell_t(X_t, A_t) + \eta\sum_{a\in\mathcal{A}}\pi_t(a|X_t)\left(g(\ell(\theta_t, X_t, a)\|\bar\ell(X_t, a)) - \frac{\mathbb{I}_{\{A_t=a\}}}{\eta\pi_t(a|X_t)}\bar\ell(X_t, a)\right.\right. \\
&\qquad \left.\left. + \frac{\mathbb{I}_{\{A_t=a\}}}{2\eta^2\pi_t(a|X_t)^2}\bar\ell(X_t, a)\right)\right] \\
&\qquad \text{(by Equation 5)} \\
&= \mathbb{E}_t\left[\eta\sum_{a\in\mathcal{A}}\pi_t(a|X_t)g(\ell(\theta_t, X_t, a)\|\bar\ell(X_t, a)) + \frac{1}{2\eta}\sum_{a\in\mathcal{A}}\bar\ell(X_t, a)\right] \\
&\qquad \text{(by the tower rule of expectation and } \mathbb{E}_t\left[\mathbb{I}_{\{A_t=a\}}\right] = \pi_t(a|X_t)) \\
&= \mathbb{E}_t\left[\eta\sum_{a\in\mathcal{A}}\pi_t(a|X_t)g(\ell(\theta^\star, X_t, a)\|\bar\ell(X_t, a)) + \frac{1}{2\eta}\sum_{a\in\mathcal{A}}\bar\ell(X_t, a)\right] \\
&\qquad \text{(using again that } \theta_t \text{ has the same conditional distribution as } \theta^\star) \\
&= \eta I_t(\theta^*; L_t) + \frac{1}{2\eta}\sum_{a\in\mathcal{A}}\mathbb{E}_t\left[\bar\ell(X_t, a)\right].
\end{aligned}
$$

Choosing the value of $\eta > 0$ for which the latter expression is minimal, we obtain $\mathbb{E}_t\left[r_t\right] \leq \sqrt{2I_t(\theta^*; L_t)\sum_{a\in\mathcal{A}}\mathbb{E}_t\left[\bar\ell_t(X_t, a)\right]}$. The proof is completed by taking the square on both sides and rearranging. $\qquad\square$

## 6  Conclusion

We have presented a new theoretical framework for analyzing Thompson sampling in contextual bandits, resulting in new results that advance the state of the art in the well-studied problem of logistic bandits. We believe that these results are encouraging and that our analytic framework may find many more applications in the future.

As always, we leave many more questions open than what we have closed. One major question regarding logistic bandits is if it is possible to improve our new results by significantly toning down the dependence on the number of actions $K$. In light of existing hardness results for nonlinear bandit problems (e.g., for generalized linear bandits with ReLU activation, Dong et al., 2021, Foster et al., 2021) we suspect that this may not be possible. As a more modest goal, we are curious to find out if the lifted information ratio can be upper bounded in terms of the smallest-slope parameter $\kappa$ as done in many other works on logistic bandits since [Filippi et al., 2010]. We conjecture that a $O(\kappa^{-2}d)$

bound on the lifted information ratio is indeed possible, but we were not able to prove it so far. This is the case for the eluder dimension [Russo and Van Roy, 2013], another complexity measure that has been used to upper-bound the regret for contextual bandits. The eluder dimension for linear losses is $O(d)$, but for nonlinear losses we know only of $O(\kappa^{-2}d)$ bounds for the generalized linear case.

More broadly, we believe that the most interesting immediate challenge is to extend our results to hold beyond the Bayesian setting. As a counterexample by Zhang [2021] shows, this may not be possible in general, but we wonder if his "feel-good" adjustment of Thompson sampling could be analyzed with the techniques we introduced in this paper.

Throughout the paper, we have studied several different settings, some of which come with a wide range of possible choices for the form of the prior and the likelihood. In some of this scenarios, updating the posterior and sampling from it may be computationally challenging. We have ignored this aspect in order to focus on the pure online-decision aspects, and implicitly assumed that posterior sampling can be performed without approximations. In practice, several heuristics have been proposed, see for instance [Dumitrascu et al., 2018] on efficient TS for logistic bandits. It would be interesting to study how approximate sampling affects our regret guarantees, along the line of [Phan et al., 2019, Mazumdar et al., 2020].

## Acknowledgments and Disclosure of Funding

This project has received funding from the European Research Council (ERC) under the European Union's Horizon 2020 research and innovation programme (Grant agreement No. 950180).

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
