# A Omitted proofs

## A.1 The proof of Lemma 4

For didactic purposes, we provide two proofs for this lemma. We first start with the simple case of distributions with finite supports that allows us to spell out the steps in the proof using simple and intuitive notation. Then, we provide a proof for general prior distributions. Some general notations that we will use throughout are the following. We let $\mathcal{D}_{\theta,x,a}$ be the distribution of the loss given context $x$, action $a$ and a fixed parameter $\theta \in \Theta$, and let $L_{\theta,x,a}$ denote the random variable with said distribution. Using this notation, notice that $L_t \sim \mathcal{D}_{\theta^*,X_t,A_t}$. Finally, we define the likelihood function $p_{\theta,t}(c) = \mathbb{P}\left[L_{\theta,X_t,A_t} = c \mid X_t, A_t\right]$

**Proof for countably supported priors.** We first assume that the support $\Theta_1 \subseteq \Theta$ is countable, which will allow us to reason about probability mass functions. In particular, with a slight abuse of our notation, we will write $Q_t(\theta) = \mathbb{P}_t\left[\theta^* = \theta\right]$ (which should otherwise be written as $Q_t(\{\theta\})$). Defining the Bayesian posterior predictive distribution $\overline{p}_t(c) = \sum_\theta Q_t(\theta) p_{\theta,t}(c)$, we can write

$$
\begin{aligned}
I_t(\theta^*; L_t) &= \mathbb{E}_t\left[\mathcal{D}_{\mathrm{KL}}\left(P_{L_t|\theta^*,\mathcal{F}_{t-1},X_t,A_t} \big\| P_{L_t|\mathcal{F}_{t-1},X_t,A_t}\right)\right] \\
&= \mathbb{E}_t\left[\mathbb{E}_{p_{\theta^*,t}}\left[\log \frac{p_{\theta^*,t}(L_t)}{\overline{p}_t(L_t)}\right]\right] = \mathbb{E}_t\left[\log \frac{p_{\theta^*,t}(L_t)}{\overline{p}_t(L_t)}\right].
\end{aligned}
\tag{6}
$$

Then, summing up and taking marginal expectations, we get

$$
\begin{aligned}
\mathbb{E}\left[\sum_{t=1}^T I_t(\theta^*; L_t)\right] &= \mathbb{E}\left[\sum_{t=1}^T \mathbb{E}_t\left[\log \frac{p_{\theta^*,t}(L_t)}{\overline{p}_t(L_t)}\right]\right] = \mathbb{E}\left[\sum_{t=1}^T \log \frac{p_{\theta^*,t}(L_t)}{\overline{p}_t(L_t)}\right] \\
&= \mathbb{E}\left[\log \frac{\prod_{t=1}^T p_{\theta^*,t}(L_t)}{\prod_{t=1}^T \overline{p}_t(L_t)}\right]
\end{aligned}
\tag{7}
$$

To proceed, let us notice that the posterior updates take the following form by definition:

$$
Q_{t+1}(\theta) = \frac{Q_t(\theta) p_{\theta,t}(L_t)}{\sum_{\theta'} Q_t(\theta') p_{\theta',t}(L_t)}.
$$

Also, let us define the notation $\overline{p}(L_{1:T}) = \sum_\theta Q_1(\theta) \prod_{t=1}^T p_{\theta,t}(L_t)$ and notice that we can express this quantity by a recursive application of the above expression as

$$
\begin{aligned}
\overline{p}(L_{1:T}) &= \prod_{t=1}^T \frac{\overline{p}(L_{1:t})}{\overline{p}(L_{1:t-1})} = \prod_{t=1}^T \frac{\sum_\theta Q_1(\theta) \prod_{k=1}^t p_{\theta,k}(L_k)}{\sum_{\theta'} Q_1(\theta') \prod_{k=1}^{t-1} p_{\theta',k}(L_k)} \\
&= \prod_{t=1}^T \sum_\theta Q_t(\theta) p_{\theta,t}(L_t) = \prod_{t=1}^T \overline{p}_t(L_t).
\end{aligned}
$$

Then, we have

$$
\log \frac{\prod_{t=1}^T p_{\theta^*,t}(L_t)}{\prod_{t=1}^T \overline{p}_t(L_t)} = \log \frac{\prod_{t=1}^T p_{\theta^*,t}(L_t)}{\overline{p}(L_{1:T})} = \log \frac{\prod_{t=1}^T p_{\theta^*,t}(L_t)}{\sum_{\theta'} Q_1(\theta') \prod_{t=1}^T p_{\theta',t}(L_t)}.
$$

$\square$

**Proof for general prior distributions.** The proof follows from similar arguments, although we cannot work with probability mass functions any more. In particular, we will denote by $Q_1$ the prior distribution of $\theta^*$, which satisfies the following identity:

$$
\mathbb{P}\left[\theta^* \in A\right] = \int_{\theta \in \Theta} \mathbb{I}_{\{\theta \in A\}} dQ_1(\theta).
$$

Similarly, we denote by $Q_{t+1}$ the posterior distribution on $\theta^*$ after round $t$, which satisfies

$$
\mathbb{P}\left[\theta^* \in A | \mathcal{F}_t\right] = \int_{\theta \in \Theta} \mathbb{I}_{\{\theta \in A\}} dQ_{t+1}(\theta)
$$

We now apply Bayes theorem for general distributions that gives the following expression for $q_{t+1}$ :

$$q_{t+1}(\theta) = \frac{q_t(\theta)p_{\theta,t}(L_t)}{\int_{\theta'\in\Theta} p_{\theta',t}(L_t)q_t(\theta')dQ_1(\theta')}$$

$$= \frac{\prod_{k=1}^{t} p_{\theta,k}(L_k)}{\int_{\theta'\in\Theta} \prod_{k=1}^{t} p_{\theta',k}(L_k)dQ_1(\theta')}$$

where $q_{t+1}(\theta) = \frac{dQ_{t+1}}{dQ_1}(\theta)$ is the Radon-Nykodim derivative of the posterior measure with respect to the prior measure which is always guaranteed to exist (cf. Theorem 1.31 of Schervish, 1996).

As in the previous proof, we once again define $\bar{p}_t(L_t) = \int_{\theta\in\Theta} p_{\theta,t}(L_t)q_t(\theta)dQ_1(\theta)$
and $\bar{p}(L_{1:T}) = \int_{\theta\in\Theta} \prod_{t=1}^{T} p_{\theta,t}(L_t)dQ_1(\theta)$, and compute the relation :

$$\bar{p}(L_{1:T}) = \prod_{t=1}^{T} \frac{\bar{p}(L_{1:t})}{\bar{p}(L_{1:t-1})} = \prod_{t=1}^{T} \frac{\int_{\theta\in\Theta} \prod_{k=1}^{t} p_{\theta,k}(L_k)dQ_1(\theta)}{\int_{\theta'\in\Theta} \prod_{k=1}^{t-1} p_{\theta',k}(L_k)dQ_1(\theta')}$$

$$= \prod_{t=1}^{T} \int_{\theta\in\Theta} q_t(\theta)p_{\theta,t}(L_t)dQ_1(\theta) = \prod_{t=1}^{T} \bar{p}_t(L_t).$$

Then, we have

$$\log \frac{\prod_{t=1}^{T} p_{\theta^*,t}(L_t)}{\prod_{t=1}^{T} \bar{p}_t(L_t)} = \log \frac{\prod_{t=1}^{T} p_{\theta^*,t}(L_t)}{\bar{p}(L_{1:T})} = \log \frac{\prod_{t=1}^{T} p_{\theta^*,t}(L_t)}{\int_{\theta'\in\Theta} \prod_{t=1}^{T} p_{\theta',t}(L_t)dQ_1(\theta')}.$$

Taking expectations and repeating the derivations of Equation (6) and Equation (7), the proof is concluded. $\quad\square$

## A.2 Proof of Theorem 2

The proof follows the same arguments as the proof of Theorem 1, except we need a different bound on the sum of information-gain terms that does not involve the entropy of $Q_1$. The bound is based on a covering argument that is provided by the following lemma, which, together with Equation (3), will directly imply the result claimed in the theorem.

**Lemma 6.** *Under the assumptions of Theorem 2,* $\mathbb{E}\left[\sum_{t=1}^{T} I_t(\theta^*; L_t)\right] \le \mathcal{N}_\varepsilon(\Theta_0, \varrho) + \varepsilon CT$.

*Proof.* For the clarity of exposition, we still assume that the support $\Theta_1$ of $Q_1$ is a countable subset of $\Theta$, even though our results can be extended to general distributions. Fix an $\varepsilon > 0$ and let $\widehat{\Theta}_\varepsilon \subset \Theta$ be a minimal $\varepsilon$-cover of $\Theta_1$. Thus, for any $\theta \in \Theta_1$, there exists $\widehat{\theta} \in \widehat{\Theta}_\varepsilon$ such that $\varrho(\theta, \widehat{\theta}) \le \varepsilon$. Let $\mathcal{G}_\varepsilon$ be the partition of $\Theta_1$ constructed by associating to each $\theta \in \Theta_1$ the closest (as measured by metric $\varrho$) element of the cover $\widehat{\theta} \in \widehat{\Theta}_\varepsilon$. For any $\theta \in \Theta_0$, we denote by $\mathcal{G}_\varepsilon(\theta)$ the unique set of the partition containing $\theta$. Note that, for any $\theta \in \Theta_1$ and $\widetilde{\theta} \in \mathcal{G}_\varepsilon(\theta)$, $\rho(\theta, \widetilde{\theta}) \le 2\varepsilon$ and so, by Assumption 1, $|\log p_{\theta,t}(L_t) - \log p_{\widetilde{\theta},t}(L_t)| \le 2C\varepsilon$. Hence, we have

$$-\log\left(\sum_{\theta\in\Theta_1} q_1(\theta)\prod_{t=1}^{T} p_{\theta,t}(L_t)\right) \le -\log\left(\sum_{\theta\in\mathcal{G}_\varepsilon(\theta^*)} q_1(\theta)\prod_{t=1}^{T} p_{\theta,t}(L_t)\right)$$

$$= -\log\left(\sum_{\theta\in\mathcal{G}_\varepsilon(\theta^*)} q_1(\theta)\prod_{t=1}^{T}\left(p_{\theta^*,t}(L_t)\cdot\frac{p_{\theta,t}(L_t)}{p_{\theta^*,t}(L_t)}\right)\right)$$

$$\le -\log\left(\sum_{\theta\in\mathcal{G}_\varepsilon(\theta^*)} q_1(\theta)\prod_{t=1}^{T}\left(p_{\theta^*,t}(L_t)\cdot e^{-2C\varepsilon}\right)\right)$$

$$\le -\log\left(\sum_{\theta\in\mathcal{G}_\varepsilon(\theta^*)} q_1(\theta)\prod_{t=1}^{T} p_{\theta^*,t}(L_t)\right) + 2C\varepsilon T$$

$$= -\log\left(\widehat{Q}_1(\mathcal{G}_\varepsilon(\theta^\star))\prod_{t=1}^{T} p_{\theta^\star,t}(L_t)\right) + 2C\varepsilon T,$$

where we denoted $\widehat{Q}_1(\mathcal{G}_\varepsilon(\theta^\star)) = \sum_{\theta\in\mathcal{G}_\varepsilon(\theta^\star)} q_1(\theta)$. Using this result together with Lemma 4, we get

$$\mathbb{E}\left[\sum_{t=1}^{T} g(\ell(\theta^\star, X_t, A_t)\|\bar{\ell}_t(X_t, A_t))\right] = \mathbb{E}\left[\log\frac{\prod_{t=1}^{T} p_{\theta^\star,t}(L_t)}{\sum_{\theta\in\Theta_1} q_1(\theta)\prod_{t=1}^{T} p_{\theta,t}(L_t)}\right]$$

$$\leq \mathbb{E}\left[\log\frac{p_{\theta^\star}(L_{1:T})}{\widehat{Q}_1(\mathcal{G}_\varepsilon(\theta^\star))p_{\theta^\star}(L_{1:T})}\right] + 2C\varepsilon T$$

$$= \mathbb{E}\left[-\log\widehat{Q}_1(\mathcal{G}_\varepsilon(\theta^\star))\right] + 2C\varepsilon T$$

$$= -\sum_{\theta\in\Theta_1} q_1(\theta)\log\widehat{Q}_1(\mathcal{G}_\varepsilon(\theta)) + 2C\varepsilon T$$

$$= -\sum_{G\in\mathcal{G}_\varepsilon} \widehat{Q}_1(G)\log\widehat{Q}_1(G) + 2C\varepsilon T$$

$$\leq \log\left(|\mathcal{G}_\varepsilon|\right) + 2C\varepsilon T.$$

The proof of the lemma is concluded by noting that $|\mathcal{G}_\varepsilon| \leq |\widehat{\Theta}_\epsilon| \leq \mathcal{N}_\varepsilon(\Theta_1, \varrho)$. $\qquad\square$

### A.3  The proofs for logistic bandits

**Proof of Theorem 4**  Using a standard result on the covering number of the Euclidean ball, we have $\mathcal{N}_\varepsilon(\Theta_0, \|\cdot\|) \leq (\frac{2S}{\varepsilon}+1)^d$. Regarding Assumption 1, notice that $\log(\sigma(f_\theta(x,a))) = f_\theta(x,a) - \log(1 + \exp(f_\theta(x,a)))$. Since $\log(\sigma(z))$ and $\log(1-\sigma(z))$ are 1-Lipschitz and $f_\theta(x,a)$ is $C$-Lipschitz, $\log P_{\theta,x,a}(1) = \log(\sigma(f_\theta(x,a)))$ and $\log P_{\theta,x,a}(0) = \log(1-\sigma(f_\theta(x,a)))$ are also $C$-Lipschitz, implying that Assumption 1 holds with the same constant $C$. The claim then follows from Theorem 2 by taking $\varepsilon = 1/CT$. $\qquad\square$

**Proof of Corollary 1**  The function $f_{\theta,x,a} : \theta \to \langle\varphi(x,a), \theta\rangle$ is linear in $\theta$ and since $\varphi(x,a)$ is bounded in norm by B, it is also $B$-Lipschitz. This proves that Assumption 1 is satisfied with the constant $B$. We can then apply Theorem 4 with $C = B$ and $\rho = 2K$ (Lemma 1). $\qquad\square$

### A.4  The Legendre–Fenchel conjugate of the binary relative entropy

**Proposition 1.** *For any $u \leq 0$ and $q \in [0, 1]$:*

$$g^*(u\|q) \leq q\left(u + \frac{u^2}{2}\right).$$

*Proof.* The claim follows from the following calculation:

$$g^*(u\|q) = \log(1 + q(e^u - 1)) \leq q(e^u - 1) \leq q\left(u + \frac{u^2}{2}\right), \tag{8}$$

where the first inequality is from $\log(1 + x) \leq x$ for any $x > -1$, and the second inequality is from $e^x \leq 1 + x + \frac{x^2}{2}$ for any $x \leq 0$. $\qquad\square$

## B  Linear bandits

By making an assumption that loss function is a linear function of some feature map over states and actions, this linear dependence allows the learner to generalize the observation among the other actions. We provide two different results for the linear bandits setting, the first one is Lemma 2 that holds for general distributions with expected loss being a linear function taking values in $(0, 1)$, and the other is Lemma 3 that works for setting where the prior of $\theta^*$ is Gaussian distribution and the observations have a Gaussian noise. These two results are connected and share a similar proof

techniques, thus we first state the following auxiliary lemma, that holds for general linear functions. We also need to define the following quantity:

$$\delta_t = \frac{\left( \mathbb{E}_t \left[ \left( \bar{\ell}_t(X_t, A_t) - \ell(\theta_t, X_t, A_t) \right) \right] \right)^2}{\mathbb{E}_t \left[ \left( \bar{\ell}_t(X_t, A_t) - \ell(\theta^*, X_t, A_t) \right)^2 \right]}. \tag{9}$$

We prove the following upper bound on this quantity:

**Lemma 7.** *Suppose the expected losses are linear functions of the form $\ell(\theta, x, a) = \langle \theta, \varphi(x, a) \rangle$, for all $x \in \mathcal{X}, a \in \mathcal{A}, \theta \in \mathbb{R}^d$, where $\varphi : \mathcal{X} \times \mathcal{A} \to \mathbb{R}^d$ is a feature map. Then, $\delta_t \le d$.*

*Proof.* The proof follows from an adaptation of Proposition 5 of Russo and Van Roy [2016]. Their technique is based on constructing a $K \times K$ matrix $M$ and showing that the information ratio can be upper bounded by the rank of this matrix. In particular, let us fix $t$ and define $\lambda_a = \pi_t(a|X_t)$ and the matrix $M$ whose elements are defined as

$$M_{a,a'} = \sqrt{\lambda_a \lambda_{a'}} \left( \bar{\ell}_t(X_t, a) - \mathbb{E}_t \left[ \ell(\theta^*, X_t, a) | A_t^* = a' \right] \right).$$

First, we relate the denominator of $\delta_t$ to this matrix as follows:

$$\mathbb{E}_t \left[ \left( \bar{\ell}_t(X_t, A_t) - \ell(\theta^*, X_t, A_t) \right)^2 \right] = \mathbb{E}_t \left[ \sum_a \pi_t(a|X_t) \left( \bar{\ell}_t(X_t, a) - \ell(\theta^*, X_t, a) \right)^2 \right]$$

$$= \sum_{a,a'} \pi_t(a|X_t) \pi_t(a'|X_t) \mathbb{E}_t \left[ \left( \bar{\ell}_t(X_t, a) - \ell(\theta^*, X_t, a) \right)^2 \Big| A_t^* = a' \right]$$

$$\ge \sum_{a,a'} \pi_t(a|X_t) \pi_t(a'|X_t) \left( \mathbb{E}_t \left[ \left( \bar{\ell}_t(X_t, a) - \ell(\theta^*, X_t, a) \right) \Big| A_t^* = a' \right] \right)^2$$

$$= \sum_{a,a'} \pi_t(a|X_t) \pi_t(a'|X_t) \left( \bar{\ell}_t(X_t, a) - \mathbb{E}_t \left[ \ell(\theta^*, X_t, a) | A_t^* = a' \right] \right)^2 = \sum_{a,a'} M_{a,a'}^2 = \|M\|_F^2,$$

where we used Jensen's inequality. For the numerator, we get

$$\mathbb{E}_t \left[ \bar{\ell}_t(X_t, A_t) - \ell(\theta^*, X_t, A_t^*) \right] = \mathbb{E}_t \left[ \bar{\ell}_t(X_t, A_t^*) - \ell(\theta^*, X_t, A_t^*) \right]$$

$$= \sum_a \pi_t(a|X_t) \left( \bar{\ell}_t(X_t, a) - \mathbb{E}_t \left[ \ell(\theta^*, X_t, a) | A_t^* = a \right] \right) = \mathrm{tr}(M).$$

Thus, we have $\left( \mathbb{E}_t \left[ \bar{\ell}_t(X_t, A_t) - \ell(\theta^*, X_t, A_t^*) \right] \right)^2 = \left( \mathrm{tr}(M) \right)^2$. Putting this together with the previous bound, we conclude that $\rho_t \le \left( \mathrm{tr}(M) / \|M\|_F \right)^2$, which can be further upper bounded by the rank of $M$. Now, one can follow the steps of the proof of Proposition 5 of Russo and Van Roy [2016] to prove that the rank of $M$ is at most $d$. $\square$

Equipped with result of Lemma 7, we can derive our analysis for the two described linear settings:

**The proof of Lemma 2** We relate the denominator of $\delta_t$, defined in (9), to the denominator of $\rho_t$, by using Pinsker's inequality as follows:

$$I_t(\theta^*; L_t) \ge 2\mathbb{E}_t \left[ \left( \bar{\ell}_t(X_t, A_t) - \ell(\theta^*, X_t, A_t) \right)^2 \right]$$

Using the definition of $\rho_t$ and Lemma 7, we get that $\rho_t \le \frac{d}{2}$. $\square$

**The proof of Lemma 3** Note that for the Gaussian likelihood, $I_t(\theta^*; L_t) = \mathbb{E}_t \left[ \left( \bar{\ell}_t(X_t, A_t) - \ell(\theta^*, X_t, A_t) \right)^2 \right]$, so we have

$$\rho_t = \frac{\left( \mathbb{E}_t \left[ \left( \bar{\ell}_t(X_t, A_t) - \ell(\theta_t, X_t, A_t) \right) \right] \right)^2}{\mathbb{E}_t \left[ \left( \bar{\ell}_t(X_t, A_t) - \ell(\theta^*, X_t, A_t) \right)^2 \right]}.$$

Lemma 7 provides $\rho_t = \delta_t \leq d$ bound. We still can tighten this results for the cases when $\log K \ll d$ by using the linearity and the closed form expression for $I_t(\theta^*; L_t)$. Under the stated assumptions, the posterior $Q_t$ is $\mathcal{N}(\overline{\theta}, \overline{\Sigma}_t)$ with $\overline{\theta}_t = \overline{\Sigma}_t^{-1} \sum_{s<t} \varphi(X_s, A_s)\varphi(X_s, A_s)^\mathsf{T} L_s$. and $\overline{\Sigma}_t = \sum_{s<t} \varphi(X_s, A_s)\varphi(X_s, A_s)^\mathsf{T} + \frac{\sigma^2}{\lambda} I$. Note that $\bar{\ell}_t(x,a) = \mathbb{E}_t[\langle \theta, \varphi(x,a)\rangle] = \langle \overline{\theta}_t, \varphi(x,a)\rangle$. We first consider the numerator of $\rho_t$. Let us define $\sigma_{t,a}^2 = \varphi(X_t, a)^\mathsf{T} \Sigma_t^{-1} \varphi(X_t, a)$, and notice that conditional on $\mathcal{F}_t, X_t$, we have $\langle \theta_t - \hat{\theta}_t, \varphi(X_t, a)\rangle \sigma_{t,a}^{-1} \sim \mathcal{N}(0,1)$. Let $Z$ denote a standard normal, then for $c > 0$, $\mathbb{P}[Z > c] \leq 0.5 e^{-c^2/2}$. Using $c_\delta = \sqrt{2\log(K/\delta)}$, we can show

$$\mathbb{P}_t\left[\left|\varphi(X_t, a)^\mathsf{T}(\theta_t - \hat{\theta}_t)\sigma_{t,a}^{-1}\right| > c_\delta \text{ for all } a \,\middle|\, X_t, A_t\right] \leq \delta.$$

Then, with probability at least $1 - \delta$,

$$(\ell_t(X_t, A_t) - \ell(\theta_t, X_t, A_t))^2 \, \sigma_{t,A_t}^{-2} \leq c_\delta.$$

Note that:

$$\mathbb{E}_t\left[(\ell_t(X_t, A_t) - \ell(\theta_t, X_t, A_t))^2 \, \sigma_{t,A_t}^{-2} \,\middle|\, X_t, A_t\right] = \frac{\mathbb{E}_t\left[(\ell_t(X_t, A_t) - \ell(\theta_t, X_t, A_t))^2 \,\middle|\, X_t, A_t\right]}{\sigma_{t,A_t}^2}.$$

In order to get the bound on this, we will use the idea of proof of Theorem 3.3 from Bubeck and Cesa-Bianchi [2012]: taking $W_t = \frac{(\ell_t(X_t,A_t) - \ell(\theta_t, X_t, A_t))^2}{2\sigma_{t,A_t}^2} - \ln K$ and using the formula

$$\mathbb{E}_t[W \,|\, X_t, A_t] \leq \int_0^1 \frac{1}{\delta} \mathbb{P}_t\left[W > \ln\frac{1}{\delta} \,\middle|\, X_t, A_t\right] d\delta,$$

we get $\mathbb{E}_t[W] \leq 1$, which gives

$$\mathbb{E}_t\left[(\ell_t(X_t, A_t) - \ell(\theta_t, X_t, A_t))^2 \, \sigma_{t,A_t}^{-2} \,\middle|\, X_t, A_t\right] \leq 2(1 + \ln K)\sigma_{t,A_t}^2.$$

Now, we notice that the denominator can be written as

$$\mathbb{E}_t\left[\left(\bar{\ell}_t(X_t, A_t) - \ell(\theta^*, X_t, A_t)\right)^2\right] = \mathbb{E}_t\left[\varphi(X_t, A_t)^\mathsf{T} \Sigma_t^{-1} \varphi(X_t, A_t)\right] = \mathbb{E}_t\left[\sigma_{t,A_t}^2\right]. \qquad (10)$$

Therefore, we can bound $\rho_t$ for all $t$ as follows:

$$\rho_t \leq \frac{\mathbb{E}_t\left[\left(\bar{\ell}_t(X_t, A_t) - \ell(\theta_t, X_t, A_t)\right)^2\right]}{\mathbb{E}_t\left[\left(\bar{\ell}_t(X_t, A_t) - \ell(\theta^*, X_t, A_t)\right)^2\right]} \leq \frac{2(1 + \ln K)\mathbb{E}_t\left[\sigma_{t,A_t}^2\right]}{\mathbb{E}_t\left[\sigma_{t,A_t}^2\right]} = 2(1 + \ln K).$$

$\qquad\qquad\qquad\qquad\qquad\qquad\qquad\qquad\qquad\qquad\qquad\qquad\qquad\qquad\qquad\qquad\qquad\qquad\qquad\qquad\qquad\square$

For the complete of our analysis, we also prove a regret bound for the Gaussian setting:

**Corollary 2.** *Consider the classic setting of linear bandits, such that the loss of playing action $a$ at time $t$ satisfies $\ell(\theta^*, x, a) \sim \mathcal{N}(\langle \theta^*, \varphi(x,a)\rangle, \sigma^2)$, where $\theta^*$ is drawn from a Gaussian prior $Q_1 = \mathcal{N}(0, \lambda I)$ with some $\lambda > 0$, $\sigma > 0$ and $\varphi : \mathcal{X} \times [K] \to \mathbb{R}^d$ is a feature map such that $\|\varphi(x,a)\| \leq B$ for all $x \in \mathcal{X}, a \in [K]$. Then, the Bayesian regret of TS satisfies*

$$\mathbb{E}[R_T] \leq \sqrt{2dT \min\{2(1 + \log K), d\} \log\left(1 + \frac{T\lambda B^2}{d\sigma^2}\right)}.$$

*Proof.* By using the form of $I_t(\theta^*; L_t)$ for the Gaussian likelihood and applying the regret decomposition of Equation (3), we get

$$\mathbb{E}[R_T] \leq \sqrt{\sum_{t=1}^T \mathbb{E}_t[\rho_t] \, \mathbb{E}\left[\sum_{t=1}^T \left(\bar{\ell}_t(X_t, A_t) - \ell(\theta^*, X_t, A_t)\right)^2\right]}. \qquad (11)$$

Then, by Lemma 3 and (10),

$$\mathbb{E}\left[R_T\right] \le \sqrt{\min\{2(1+\ln K), d\} T \sum_{t=1}^{T} \mathbb{E}_t \left[\varphi(X_t, A_t)^\intercal \Sigma_t^{-1} \varphi(X_t, A_t)\right]}$$

The term $\sum_{t=1}^{T} \mathbb{E}_t \left[\varphi(X_t, A_t)^\intercal \Sigma_t^{-1} \varphi(X_t, A_t)\right]$ can be addressed by using the elliptical potential lemma (e.g. Lemma 19.4. in Lattimore and Szepesvári [2020]), which is widely used in the analysis of linear bandits. Applying this lemma, we get

$$\sum_{t=1}^{T} \varphi(X_t, A_t)^\intercal \Sigma_t^{-1} \varphi(X_t, A_t) \le 2d \log\left(1 + \frac{T\lambda B^2}{d\sigma^2}\right),$$

which completes the proof. □