# OpenReview forum: "Lifting the Information Ratio: An Information-Theoretic Analysis of Thompson Sampling for Contextual  Bandits"
_NeurIPS.cc/2022/Conference — NeurIPS 2022 Accept_

### Official Review · Reviewer_piBD · 2022-07-11

**Rating:** 7
**Confidence:** 3
**Soundness:** 3 good
**Presentation:** 3 good
**Contribution:** 3 good

**Summary:**

Previously, information ratio was used to bound regret of Thomspon sampling algorithm for multi-armed bandits. This paper provides examples that information ratio loses its usefulness in deriving bounds for contextual bandits. They introduce a new lifted information ratio which deals with information gained about parameter instead of best action. Using this ratio, the authors derive regret bounds for Thomspon Sampling. Using the bounds, they derive regret bounds under several interesting cases.

**Questions:**

Paper can benefit more form discussion on tractability.

**Limitations:**

Satisfied with authors responses.

**Strengths And Weaknesses:**

Strengths:
Lifted information ratio is an intuitive yet interesting quantity which authors motivate quite well. They also provide tight regret bounds using the new quantity.

Weaknesses:
As paper mentions, they don't provide any tractable algorithms and leave that task for future. It would be better if they can add more discussion on this.

---

> ### Author Response · Authors · 2022-07-31
> **response**
>
> Thank you for your positive evaluation of our paper!
>
> Regarding tractability: Rather than a future extension, we see this as an orthogonal problem that is already an active area of study. We have only mentioned these aspects in the conclusion to highlight that we are, in fact, not addressing these concerns in this work and we focus purely on the statistical aspects of TS. This is very much in line with virtually all the related work we cite in the paper (which are all very impactful papers despite sharing the same limitations). As such, we don’t see the lack of discussion of computational aspects to be a limitation of our work, and we will make this point more clear in the final version.

---

### Official Review · Reviewer_hEwR · 2022-07-11

**Rating:** 6
**Confidence:** 3
**Soundness:** 3 good
**Presentation:** 4 excellent
**Contribution:** 2 fair

**Summary:**

The authors study the Bayesian regret of Thompson sampling with binary losses and adversarial contexts. Adapting the information theoretic approach from Russo and Van Roy, through a definition of lifted information ratio, they provide a regret bound similar to the non-contextual case. This leads to state-of-the-art regret guarantees in many well-known cases covered by the proposed setting. Notably, they prove a regret bound for K actions, and Lipschitz logits which is independent on the slope of the link function.

**Questions:**

- Please look at the weakness part. Addressing the first two points will improve the paper.


**Limitations:**

Being a theoretical work any adverse social impact is hard to foresee.

**Strengths And Weaknesses:**

Strengths
- The lifting of information ratio is supported by the intuition that the information of the optimal action in one context may be useless in the other context. Thus substituting the mutual information between optimal action, and taken action and loss, with the mutual information between true parameter, and the loss resolves the problem.
- The adaptation of the information theoretic framework for Bayesian regret bounds in adversarial contextual setting is new. This has the potential for generalization in many other settings where information theoretic framework is used.
- The current approach results in state-of-art regret guarantees in many problems with adversarial contexts.
- The relation to the “decoupling coefficient” proposed by Zhang is also interesting (but this raises some question as well).


Weaknesses
- Moving away from the mutual information between optimal action to true parameter suggests we may end up with worse scaling w.r.t. the number of actions. However, that does not seem to be the case as seen in Lemma 3 which provides a $log(K)$ regret bound for $K$ actions. Maybe the authors should expand on how that is achieved.

- Although the intuition supports the lifted Information ratio approach, it will be good to more rigorously point out why the previous information ratio fails in the contextual case.

- Zhang explicitly mentions that TS may not be efficient in some cases, and introduces a bias towards higher reward in the Feel Good TS  approach. In light of that, the current connection seems to suggest TS suffice, at least for Gaussian case. Is this special for Gaussian case, and will fail in more general situations?


- Can there be an approach where we can judicially choose between the mutual information between optimal action, vs the true parameter while analyzing TS? It will be great if such an approach is found, as we can incorporate that in the analysis. Some comments will be interesting.

Post rebuttal:
- The current paper seems like a special case of the the notion of "learning target" discussed in https://arxiv.org/abs/2103.04047 and https://arxiv.org/abs/2201.01902 (as pointed out by another Reviewer).

---

> ### Author Response · Authors · 2022-07-31
> **response**
>
> Thank you for your thorough reading of our paper and supportive comments! We address all of your questions raised in the "weakness" section below.
>
> > Moving away from the mutual information between optimal action to true parameter suggests we may end up with worse scaling w.r.t. the number of actions. However, that does not seem to be the case as seen in Lemma 3 which provides a $\log K$ regret bound for $K$ actions. Maybe the authors should expand on how that is achieved.
>
> There are several ways in which our move from the standard information ratio to the lifted version impact the scaling of the bounds with respect to $K$. As our arguments show, the information-ratio terms are largely unaffected by the change, and the only terms that can substantially increase are these entropies. The bounds depending on the entropy $H(\theta^*)$ instead of $H(A^*)$ or $H(\pi^*)$ may indeed suggest a worse dependence on the number of actions, e.g., when the action space is finite but the outcome space is infinite. In the case of Gaussian parameters, these terms can be controlled reasonably, as the proof of Lemma 3 demonstrates. We will clarify the contribution of each of these factors more explicitly in the final version.
>
> > Although the intuition supports the lifted Information ratio approach, it will be good to more rigorously point out why the previous information ratio fails in the contextual case.
>
> Concepts of information ratio for the contextual setting have indeed been suggested in previous work on information-directed sampling (https://arxiv.org/abs/2002.11182, https://arxiv.org/abs/2205.10895). Some potential flaws of information gain about the optimal action have already been pointed out in these works (see, e.g., page 9 of https://arxiv.org/abs/2002.11182). Note however that their analysis heavily relies on the assumptions that the contexts are i.i.d., while we consider the more challenging setting of adversarial contexts. We have included a discussion of these and other related works in a revised version of the paper (changes in red). Intuitively, previous approaches fail because they try to infer the optimal actions instead of the environment parameters.
>
> > Zhang explicitly mentions that TS may not be efficient in some cases, and introduces a bias towards higher reward in the Feel Good TS approach. In light of that, the current connection seems to suggest TS suffice, at least for Gaussian case. Is this special for Gaussian case, and will fail in more general situations?
>
> Explicit optimism is indeed necessary (to the best of our knowledge) in the frequentist setting, where $\theta^*$ is arbitrarily fixed by the environment. In this paper, we consider the Bayesian setting, where we assume $\theta^*$ is really sampled from the prior, and the regret we bound is in expectation w.r.t. $\theta^*$. In this setting, TS is sufficient, as also observed by Zhang.
>
> > Can there be an approach where we can judicially choose between the mutual information between optimal action, vs the true parameter while analyzing TS? It will be great if such an approach is found, as we can incorporate that in the analysis. Some comments will be interesting.
>
> This is a very interesting question! At the moment, we believe that the choice of the “learning target” (parameter vs. optimal action) is always going to be up for the one conducting the analysis, and it seems very challenging to come up with a “universal analysis” that is adaptive to this target. Many such questions are discussed in the following paper that have learned about very recently: https://arxiv.org/abs/2103.04047. The lucky thing with TS is that these dilemmas only arise in the analysis, but not in the implementation of the algorithm. This is to be contrasted with other algorithms like information-directed sampling or UCB — see, for instance the “best of both worlds” approach of https://arxiv.org/abs/1905.10040, where the agent can detect whether it’s confronting a multi-armed or a linear contextual bandit and play the most appropriate algorithm.

---

> > ### Comment · Reviewer_hEwR · 2022-08-04
> > **Response to the Rebuttal keeping in light of the review of Reviewer qsXa**
> >
> > I thank the authors for their response. The concerns raised by me is addressed properly. However, in light of the concern raised by Reviewer qsXa, the contributions of this paper seems to be lesser compared to my initial assessment.
> >
> > I have read the response of the authors to the following concern, and it seems to hinge upon a clearer presentation of a special case of the mentioned prior works. This is a much limited claim than the one made in the original paper.
> >
> > > .. at an abstract level the lifted information ratio seems to essentially be a special case of the notion of "learning target" discussed in https://arxiv.org/abs/2103.04047 and https://arxiv.org/abs/2201.01902.

---

> > > ### Author Response · Authors · 2022-08-08
> > > **thank you**
> > >
> > > Thank you for getting back to us!
> > > We of course understand why you decided to lower your score --- it makes perfect sense given all the relevant literature we missed in the original version. We would appreciate though if you could take a quick look at the revision and confirm that the discussion of these related works is now fair. It appears that the reviewer who originally raised these concerns is happy with our response, and we hope that you share this view with them. In any case, we are happy to see that you are still leaning towards recommending acceptance.
> > > Once again, thank you for your work & best regards,
> > > the authors

---

### Official Review · Reviewer_qsXa · 2022-07-11

**Rating:** 6
**Confidence:** 4
**Soundness:** 4 excellent
**Presentation:** 3 good
**Contribution:** 2 fair

**Summary:**

This paper focuses on parametrized contextual bandit problems, and their analysis via the Russo-Van Roy information ratio framework. The main contribution is to observe that one can analyze information gain on the "lifted" level of the "underlying context" instead of the "optimal action". This leads to nice new regret bounds for logistic bandits. The analyses are clean and remove dependence on problem-dependent constants which were common in previous work. At a high level, it seems that the "lifted" information ratio may often be the right way to study parametrized contextual bandits.

**Questions:**

The most pressing suggestion is to incorporate the prior work on general learning targets. Assuming I'm not mistaken about the strong similarity, I suggest the authors make substantial revisions to acknowledge, compare with, and perhaps build upon this earlier work. For example, introducing the term "lifted information ratio" seems inappropriate if the idea already has a name. I think the submitted paper still has plenty of merit, and it seems plausible to me that the application to logistic bandits gives a more convincing application of this idea than the previous works I mentioned. I'm not familiar enough with the literature to say. If I'm confused about this point, please let me know and I will be happy to raise my score.

I have a few other questions and comments:

1. The authors write:

"When considering i.i.d. contexts and finite policy classes, one can apply the theory of Russo and Van Roy [2016] and treat policies as actions to obtain regret bounds scaling with the entropy of the optimal policy, but this can lead to a polynomial dependence on the number of contexts."

I would appreciate some elaboration on this point. My understanding is that the authors have in mind the setting of Thompson sampling with changing feedback graphs G_t (which for contextual bandit would be a union of K vertex-disjoint cliques). I believe the regret bounds here are of order \sqrt{TK H} where H is the entropy of the optimal policy (e.g. https://arxiv.org/abs/1805.08930 or section 8 of https://arxiv.org/abs/1902.00681). I don't see this polynomial dependence on the number of contexts anywhere and would like some clarification. It's quite possible that I am misunderstanding something basic and I apologize if this is the case.

2. In line 101, I was a bit surprised that the adversary does not know \theta^*. It would be nice to have some commentary on whether this is likely to be truly essential, when it is used, if it is required in related work, etc. On its face it seems to detract a little bit from the picture.

3. At the end of page 7, I was confused about the Bayesian telescoping argument. In particular it looks to me like the usual entropy chain rule should suffice: the expected information-gain at time t is by definition the average decrease in the entropy of \theta^*. Summing and using the fact that entropy is always non-negative, the upper bound of H(\theta^*) should follow. So it seems like the calculation was overcomplicated a bit and I cannot tell why.

4. Equation (1): missing a comma in the first \ell(\cdot) expression.

5. A positive comment to finish: I appreciated that the authors focused on countable priors while making it clear that this is just for convenience. I thought this choice made things quite pleasant to read.



**Strengths And Weaknesses:**

I like the results and the writing is very nice.

As far as I can tell, the main problem is that the key idea is less novel than the authors seem to believe. In particular while the application to logistic bandits is probably new, at an abstract level the lifted information ratio seems to essentially be a special case of the notion of "learning target" discussed in https://arxiv.org/abs/2103.04047 and https://arxiv.org/abs/2201.01902. (The adversarial contexts may change things slightly, but it seems mostly inconsequential for my comment.) I expand on what to do about this in the "Questions" section. In short, I think the submission should probably be revised to take this literature into account, and then resubmitted to a future conference (where I would support its acceptance).

---

> ### Author Response · Authors · 2022-07-31
> **revision**
>
> Thank you for your deep reading of our paper and insightful comments! We address all concerns below.
>
> ## Missing references
>
> Thank you very much for calling these references into our attention! These omissions are indeed embarrassing, and we wish to apologize for messing up the literature review this badly. In fact, we have learned about these works shortly after submission, and also about some other recent works that used very similar concepts of information gain and information ratio: https://arxiv.org/pdf/2002.11182.pdf and https://arxiv.org/abs/2205.10895 (note that the second one appeared online after the NeurIPS deadline).
>
> All that said, we still believe that our analytic framework that clearly highlights the role of the “lifted information ratio” is novel, at least in the sense of framing the key elements of the analysis in a crisp and transparent way, which is potentially interesting for the NeurIPS audience and can inspire future work. Our results on logistic bandits are novel and arguably significant, which clearly demonstrates the usefulness of the framework. In this sense, we strongly believe that the paper is suitable for publication, after all the necessary adjustments are made on the presentation side.
>
> We are fully committed to updating the paper to take these related works into account and position our contribution more carefully within the related literature. In defense of our terminology, we believe that the term “lifted information ratio” is illuminating, and find the practice of calling all forms of information ratio by the same name to be unnecessarily confusing. Thus, we wish to keep this terminology for the final version of the paper, although we will significantly tone down our claims about the novelty of this concept. To demonstrate our commitment, we have updated the paper to take these works into account and highlighted the changes in red. (For now, we have kept the changes minimal to keep the load on the reviewers’ as low as possible. We will make further changes in the final version along the same lines to make the entire narrative of the paper fully consistent.)
>
> ## Other comments
>
> 1. We have updated our discussion on a potential polynomial scaling with the number of contexts --- what we meant is that in very the worst case, the entropy of the policies can scale very poorly with the number of contexts. Our bounds using covering arguments get around this poor scaling under some assumptions on the loss function.
> 2. The contexts being independent of the choice of $\theta^*$ is a standard assumption (see e.g. https://arxiv.org/pdf/1209.3352.pdf). The setting where the adversary knows $\theta^*$ can lead to subtle measurability issues in our analysis, and we preferred to stay away from these complications in this paper. We will add a pointer to previous work to justify the assumption.
> 3. It was our choice to state the proofs in terms of elementary algebra rather than using the algebra of information theory. We think this improves clarity and makes the argument more convincing. At least we ourselves have found this reasoning to be much more transparent than the standard information-theoretic argument, and also less prone to errors that can arise due to conditioning the mutual-information terms on the non-i.i.d. process $(X_t,A_t,\dots,X_1,A_1)$. We are confident that some readers sharing our background will also find this type of reasoning to be easier to follow.
>
> ## Summary
>
> We would like to thank you once again for your thoughtful and supportive review. We hope that you find our response above to be convincing evidence that we are willing and able to revise the paper in a way that acknowledges previous work sufficiently without making large uncheckable changes to the original content. We would be happy to resolve any further doubts you may have.

---

> > ### Comment · Reviewer_qsXa · 2022-08-04
> > **Thank you for the reply and revision**
> >
> > Thanks for addressing my comments. The revision looks to be in the right direction and I raised my score. (I actually didn't realize that submitting revisions post-reviewing was possible.) As they stand, the results are quite nice, especially the removal of annoying dependence on the link functions.

---

> > > ### Author Response · Authors · 2022-08-08
> > > **thank you!**
> > >
> > > Thank you very much for checking the revision on such a short notice and revising your score --- we really appreciate your work!
> > > Best regards,
> > > the authors

---

### Meta-Review · Area_Chair_xAEX · 2022-08-26

**Recommendation:** Accept
**Confidence:** Certain

**Metareview:**

The paper presents an analysis of the Bayesian regret of Thompson Sampling algorithm in contextual bandits under an adversarial context process. The authors express the regret as a function of the so-called lifted information ratio. This information theoretical quantity is a natural extension of those introduced by Russo and Van Roy. The analysis provides new regret upper bounds for logistic bandit, and comes with simpler and elegant proofs.

The reviewers easily reached a consensus on this paper. It is very well written, easy to follow, and provide very interesting contributions.

The discussion phase gave the opportunity for the authors to correct the related work section, and in particular to re-position the paper compared to papers pointed out by during the review process.


**Award:**

No

---

### Decision · Program_Chairs · 2022-09-14

Accept